



# Nonparametric lower bounds to mean transit times

Earl Bardsley[1]

[1]School of Science, University of Waikato, Hamilton 3240, New Zealand

*Correspondence to*: Earl Bardsley (earl.bardsley@waikato.ac.nz)

**Abstract.** Mean transit time $\mu_T$, also called mean residence time, has been used widely in hydrological studies as an indicator of catchment water storage characteristics. Typically $\mu_T$ is estimated by the nature of catchment transformation of a natural input tracer time series. For example, increased damping and delaying of $^{18}O$ seasonal isotopic variation may be taken to indicate longer mean transit times. Part of a $\mu_T$ estimation process involves specification of a lumped parameter flow

model which provides the basis for a parametric transit time distribution. However, $\mu_T$ estimation has been called into question because catchment flow systems have a degree of complexity which may not justify use of simple parametric distributions. Moving toward a related index, the question is raised here as to the extent to which an arbitrary transit time distribution might enable a model mean transit time to be minimized before the fit to catchment output tracer data becomes unacceptably poor. This minimized mean value $\mu_*$ represents a lower bound to $\mu_T$, whatever the true transit time distribution

might be. The lower bound is not necessarily an approximation to $\mu_T$ but might serve as an index for catchment comparisons or detect when $\mu_T$ is large. For a linear catchment system a simple nonparametric linear programming (LP) approach can be utilised to obtain $\mu_*$, which is conditional on a user-specified acceptable level of data fit. The LP method presented is applicable to both steady state and time-varying catchment systems and has the advantage of not requiring specification of lumped parameter models or use of explicit transit time distributions.

## 1 Introduction

Kirchner (2016a) noted that the variability of natural catchments calls into question the simple tracer transit time parametric distributions frequently assumed in the literature. This is because transit times of natural tracer such as $^{18}O$ must be in the form of complex and unknown mixed probability distributions, reflecting many different component distributions arising from a large number of different spatial input points and flow paths to a recording site.


As emphasised by Kirchner (2016a), applying simple lumped parameter catchment models to a complex reality may lead to incorrect conclusions with respect to mean transit times. In this regard it could be noted the widespread use of the well-mixed model (exponential distribution of transit times) seems particularly inappropriate because the assumption is that in a given small interval of time all tracer particles must have equal probability of passing out the catchment exit. This must

apply regardless of tracer particle location on the land surface or below ground. A physical situation of this type will never





hold in a groundwater flow system and only approximately in a catchment for the special case where a shallow lake occupies most of the area. Similarly, gamma distributions do not warrant special consideration as transit-time distributions outside of idealised situations, except for the special case where the catchment largely comprises a cascade of well-mixed lakes (gamma distributions obtained as sums of independent exponential random variables). Generally speaking, no probability

distribution can claim particular theoretical justification in any form of hydrological study unless the situation of the physical environment matches the statistical characterization of the distribution concerned. Such characterization of course excludes fortuitous empirical matching of a given distribution to recorded or simulated data.

In view of errors which may arise with the use of the mean transit time as a catchment storage metric, Kirchner (2016a,b)
proposed that focus should be on the younger portion of the transit time distribution. Specifically, the fraction of young water (< 0.2 years) at exit was suggested as a better alternative metric to achieve from seasonal amplitude comparisons between input and output tracer time series. A case was made also that this metric will be robust against non-stationarity and nonlinear effects.

However, as was also noted by Kirchner (2016a), mean transit time has a clear meaning and long history of application in hydrological studies. It might therefore be worth having a related index which maintains a degree of connection with mean transit time. This technical note outlines a simple linear programming (LP) methodology to give mean transit time lower bounds, which might serve as metrics for catchments which can be approximated as linear systems. The LP method is applicable to both steady state and time-varying linear systems.

**2 Definitions**

As part of catchment precipitation there is time-varying input of a passive tracer. Some of the tracer exits via river discharge at a recording site at the lower end of the catchment and some (depending on the tracer) may depart by evaporation loss. The assumption is made that at time of entry into the catchment, all particles of a given tracer type will have the same time-invariant probability of departing at the recording site.


Define two data time series $X_1, X_2, \dots X_K$ and $Y_1, Y_2, \dots Y_K$ to be flux-weighted catchment tracer inputs and recording site outputs, respectively, with all values being time averages for consecutive equal time intervals $\Delta t_1, \Delta t_2 .. \Delta t_K$. In addition, there are further $X$ values available for $X_0, X_{-1}, X_{-2} \dots X_{1-N}$.

For a tracer particle exiting at the recording site, its transit time $T$ is a random variable corresponding to the time spent resident within the catchment between entry and exit. The probability distribution for $T$ is the transit time distribution with mean transit time $\mu_T$. If $\mu_T$ remains constant over time then it is a hydrological parameter of that catchment.

Transit time distributions are usually represented in models as known, continuous, and unbounded to the right. However, the approach adopted here is to use transit time distributions defined as being always unknown, right-bounded, and discrete. That is, a transit time distribution with unknown mean $\mu$ is represented as the unknown probability mass function $P(\tau)$, defined over the integer times $\tau = 0, 1, 2, .. N$. The choice of $N$ is not critical as long as it is sufficiently large not to have influence later when seeking a lower bound for $\mu_T$. The inclusion of $\tau = 0$ may seem unusual because it implies some tracer being instantaneously transported to the recording site. A finite probability of zero time is of practical value, however, because it ensures that any calculated lower bound to $\mu_T$ is not slightly higher than need be, as would be the case if all transit times were bounded below at 1.0.

As noted by McCallum et al., (2014) the advantage of having an arbitrary histogram as a transit time distribution is that there is no requirement to select a lumped parameter catchment model with specified parameter values. However, a negative aspect is that the flexibility of an arbitrary transit time distribution also creates a highly undetermined situation with respect to distribution definition (Visser, et al., 2013). That is, there may be many different $P(\tau)$ configurations with different mean values that lead to equally good matching of the $Y$ data.

Identifiability of $P(\tau)$ is not a specific issue in the present context, however, because $P(\tau)$ does not have to be estimated to achieve a lower bound to $\mu_T$, with $P(\tau)$ maintaining full flexibility other than summation to 1.0. It follows that configuring $P(\tau)$ to minimize $\mu$ down to $\mu_*$ can serve as a means to obtain a lower bound to $\mu_T$, subject to maintaining a user-defined $Y$ data fit. That is, the lower bound $\mu_*$ is obtained without knowledge of the true transit time distribution.

Given a catchment which has a flow system which is linear to a first approximation (linear superposition of tracer), a simple LP approach can be applied to carry out the constrained minimization. Two linear situations are considered. Section 3 is concerned with LP minimization for the steady state case, which by definition has a single time-invariant transit time distribution. Section 4 describes LP minimization for time-varying transit time distributions for the special case where the distributions are different but all have the same mean value $\mu$. In all LP minimizations the variables involved are non-negative, not explicitly re-stated here in the LP specifications.

### 3 Lower-bounding of $\mu_T$ (steady state)

For a steady state situation the time-invariant transit time distribution is represented by the unknown probability mass function $P(\tau)$ with unknown mean value $\mu$.

At a given time $t$ the model-predicted tracer output $\hat{Y}_t$ at the recording site can be expressed by a discrete equivalent of the convolution integral:



$$\hat{Y}_t \;=\; \theta \sum_{\tau=0}^{N} X_{t-\tau} P(\tau) \qquad = \quad \sum_{\tau=0}^{N} X_{t-\tau}\, \omega_\tau \qquad\qquad (1)$$

5    where θ is a scale parameter reflecting the ratio of the means of the $X$ and $Y$ tracer time series, and $\omega_t = \theta\, P(\tau)$.

The first step is to carry out LP minimisation which fits the linear model of Eq. (1) to the $K$ values of $Y_t$. The LP process

adjusts the $\omega_\tau$ values to obtain the minimum value of mean absolute deviations between the observed and model-predicted

values:

$$\text{MINIMIZE} \quad K^{-1} \sum_{t=1}^{K} \left| Y_t - \sum_{\tau=0}^{N} X_{t-\tau}\, \omega_\tau \right| \qquad\qquad (2)$$

The scale parameter θ is then obtained as:

$$\theta = \sum_{\tau=0}^{N} \omega_\tau \qquad\qquad (3)$$

and $P(\tau) = \omega_\tau / \theta$, for $\tau = 0, 1, \ldots N$.

20

If it happens that the best fit mean absolute deviation $\bar{D}_{min}$ gives an unacceptably poor match to the $Y$ values then it is not

possible to obtain a lower bound to $\mu_T$ with a steady state model. A time-varying linear model as described in Section 4

might then be considered.

25    Given an acceptable or perhaps even perfect match to the $Y$ data from LP fitting, the resulting individual $P(\tau)$ probabilities

will most likely be poorly defined, as noted earlier. That is, there may be many different probability configurations of $P(\tau)$

which yield a similar degree of fit to the $Y$ values.





Next, let $\bar{D}_*$ symbolise a user-specified upper bound to the mean of the absolute deviations from data in an LP minimization of μ, where $\bar{D}_* > \bar{D}_{\min}$.

For a given $\bar{D}_*$, a constrained LP minimization of μ can be carried out by adjusting the $P(\tau)$ probabilities:

MINIMIZE $\qquad \displaystyle\sum_{\tau=0}^{N} \tau P(\tau)$ $\hspace{4cm}$ (4)

subject to

$$K^{-1}\sum_{t=1}^{K}\left|Y_t - \theta\sum_{\tau=0}^{N} X_{t-\tau} P(\tau)\right| \;\leq\; \bar{D}_*, \qquad \sum_{\tau=0}^{N} P(\tau) = 1$$

LP minimisations of μ are carried out for different values of $\bar{D}_*$, with $\bar{D}_*$ being incremented upwards from some initial value close to $\bar{D}_{\min}$, taking note of $\bar{D}_*$ and μ* each time, where μ* is the minimized value of μ in any instance. There need not be a

large number of minimisations because it is only necessary to establish an approximate plot of μ* against $\bar{D}_*$, giving a graphical indication of the sensitivity of μ* to data fit.

The value of μ* will decrease as $\bar{D}_*$ is increased. The point of interest is when $\bar{D}_*$ is sufficiently large that the poor fit can be taken as indicative that μ_T must be greater than that μ*, given a steady state condition. The lower bound to μ_T is thus defined

as the value of μ* associated with the just-acceptable degree of data fit as measured by $\bar{D}_*$. There is therefore a degree of subjectivity with the lower bound because perceptions may differ over what constitutes a just-acceptable fit. Nonetheless, plots of μ* against $\bar{D}_*$ give useful summaries of the strength of evidence for a given lower bound, which in principle can be compared from one catchment to the next.

The LP minimization operation of Eq. (4) may yield $P(\tau)$ values which are highly irregular and unlike any plausible transit time distribution. However, these probabilities are not an estimate of the true transit time distribution but just arise as the $P(\tau)$ set which happens to give the minimum possible value of μ. It may sometimes be possible to achieve slightly greater μ* values by imposing some constraints on the $P(\tau)$ probabilities. For example, a set of binary variables could be introduced to prevent multiple modes in $P(\tau)$, if that was justifiable for the catchment concerned. Another approach which has sometimes

used with nonparametric transit time distributions is to introduce a degree of smoothing (Cirpka et al., 2007; Liao et al., 2014).





## 4 Lower-bounding of $\mu_T$ (time-varying)

A similar LP approach to lower bound inference on $\mu_T$ can be taken for linear systems which are time-varying. This additional flexibility may come closer to catchment behaviour because there is accumulated evidence for transit time distributions being non-stationary – see, for example, Kirchner (2016a) and cited references. However, with a linear model there is inevitably a degree of approximation involved because in the model each cohort of tracer input is free to move differently and independently of every other cohort.

One specific time-varying model is considered here, with each $\Delta t$ initiating a different $P(\tau)$, but all $P(\tau)$ distributions are constrained to have a common unknown time-invariant mean $\mu$.

Because the different $P(\tau)$ distributions have different time origins, a given discrete distribution is now symbolised as $P_\varepsilon(.)$, which has the meaning that the distribution concerned is defined over the time values $t = \varepsilon$, $t = \varepsilon + 1 \dots t = \varepsilon + N$. At a given time $t$ the model-predicted tracer output $\hat{Y}_t$ at the recording site is given by:

$$\hat{Y}_t = \theta \sum_{\tau=0}^{N} X_{t-\tau} P_{t-\tau}(\tau) = \sum_{\tau=0}^{N} X_{t-\tau} \omega_{t-\tau} \tag{5}$$

The data fit minimization expression giving the least mean absolute deviation from the $Y$ data is therefore:

$$\text{MINIMIZE} \quad K^{-1} \sum_{t=1}^{K} \left| Y_t - \sum_{\tau=0}^{N} X_{t-\tau} \omega_{t-\tau} \right| \tag{6}$$

subject to the equality constraints:

$$\sum_{\tau=0}^{N} \omega_{1-\tau} = \theta \quad , \quad \sum_{\tau=0}^{N} \omega_{2-\tau} = \theta \quad , \quad \dots \sum_{\tau=0}^{N} \omega_{K-\tau} = \theta$$

$$\sum_{\tau=0}^{N} \omega_{K-\tau-1} = \theta \ , \quad \sum_{\tau=0}^{N} \omega_{K-\tau-2} = \theta \ , \ \dots \quad \sum_{\tau=0}^{N} \omega_{K} = \theta$$



where the scale parameter θ is obtained from the minimization and the individual $P_\varepsilon(\tau)$ probabilities are found by rescaling with θ as before. If an acceptable data fit cannot be achieved at this point then a more linear flexible model could be evaluated, allowing some degree of variation of the respective μ values.

In the data-fitting minimization of Eq. (6) there is implication that some of the discrete distribution probabilities fall outside the recorded $Y_t$ time range of $t = 1, 2, .. K$ and are therefore not fully determined at best fit by the data.

Given an acceptable data fit from the minimization of Eq. (6), constrained LP minimization of μ can be carried out for a specified value of data fit $\bar{D}_*$, with the minimization operation adjusting the probability values of all the discrete arrival time
distributions involved:

MINIMIZE    $$\sum_{t=-N}^{K} \sum_{\tau=0}^{N} \tau P_t(\tau)$$    (7)

subject to :

$$\sum_{\tau=0}^{N} P_{-N}(\tau) = 1 \quad , \quad \sum_{\tau=0}^{N} \tau P_{-N}(\tau) = \mu$$

$$\sum_{\tau=0}^{N} P_{-N+1}(\tau) = 1 \quad , \quad \sum_{\tau=0}^{N} \tau P_{-N+1}(\tau) = \mu$$

. . . .

$$\sum_{\tau=0}^{N} P_K(\tau) = 1 \quad , \quad \sum_{\tau=0}^{N} \tau P_K(\tau) = \mu$$

and

$$K^{-1} \sum_{t=1}^{K} \left| Y_t - \theta \sum_{\tau=0}^{N} X_{t-\tau} P_{t-\tau}(\tau) \right| \leq \bar{D}_*$$

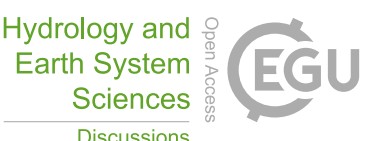

The minimization given by Eq. (7) yields the smallest possible $\mu$ value ($= \mu_*$), conditional on the goodness of fit being no worse than the specified $\bar{D}_*$. As before, minimizations are carried out over a range of $\bar{D}_*$ values to create a plot of $\mu_*$ against goodness of fit, which will show the sensitivity of any inferred lower bound to $\mu_T$ as a function of the degree of data fit.

It is known that catchment space and time scales and weather variations may all result in non-stationarity of transit time means as well as non-stationarity distribution forms (Selle et al. 2015; Peralta-Tapia et al. 2016). Modelling to allow for $\mu_T$ to be non-stationarity requires removal of the $\mu$ equality constraints in the minimization expression given by Eq. (7). However, complete freedom of variation for the many mean values would not be helpful because the LP minimisation may lead to situations far removed from reality where some $\mu$ values are set to zero, even though there is tracer input. Some linear
constraints on the range of permitted variation of the $\mu$ values would therefore need to be imposed on the basis of local experience.

## 5 Illustration

The LP method is illustrated here for the steady state condition by using a simple synthetic tracer input time series
incorporating a 12-month cycle of monthly means as a simple square wave with six values of 5.0 followed by 6 values of 10.0 (arbitrary units). This simulation was extended over 21 consecutive cycles. The $X$ data set was created by adding a uniform random variable (distributed over $0 \pm 3$) to each value in the sequence.

Simulation of the $Y$ data sets from $X$ requires specification of transit time distributions and the gamma probability density
function was utilised here, with parameterisation:

$$f(x) = \{\Gamma(\alpha)\beta^{\alpha}\}^{-1} x^{\alpha-1} e^{-x/\beta} \qquad (8)$$

where $\alpha$ is a shape parameter, $\beta$ is a scale parameter, and the mean value is $\alpha\beta$. Two gamma transit time distributions were
selected, with parameter combinations $\alpha = 5$, $\beta = 1.2$ and $\alpha = 5$, $\beta = 2.4$, respectively. The gamma scale parameter $\alpha$ is specified as measured in months, giving respective $\mu_T$ values of 6 and 12 months. The two distributions have the same positively skewed unimodal form and differ only in scale (Fig. 1).

The two gamma transit time distributions were applied to the synthetic $X$ data values to give the $Y$ recording site simulated
data, with $\mu_T = 12$ months giving the expected greater delay and damping of the input values (Fig. 2). The advantage of using simulated data of this type is that $\mu_T$ is known exactly and it is then of interest to apply the nonparametric approach to seek lower bounds for those means without reference to the gamma distributions.





For the purpose of modelling, a nonparametric discrete arrival time distribution $P(\tau)$ with $N = 23$ was employed over month integers $\tau = 0, 1, 2 \ldots 23$.

Considering first the case of $\mu_T = 6$ months, the gamma distribution concerned was discovered quite accurately as $P(\tau)$ from the LP data fitting process (Fig. 3a), with the associated $\mu$ being almost exactly 6.0 months. As expected, the model scale parameter $\theta$ was obtained as 1.0. However, the gamma discovery is an artefact of the perfect linear data simulation process with gamma distribution convolution and there is no suggestion that data fitting with a nonparametric histogram will generally result in a useful estimate of a transit time distribution or its mean value.

A similar result was obtained for LP data fitting to the $Y$ data which derived from the gamma distribution convolution with $\mu_T = 12$ months. Again , $\theta$ was obtained as 1.0 but because the $P(\tau)$ distribution extends only to $\tau = 23$ months there is a truncation effect in the distribution upper tail in this case (Fig. 3b). After data fitting this results in a slightly lower mean transit time for $P(\tau)$, giving $\mu = 11.4$ months rather than the true $\mu_T$ of 12 months. However, as shown later, this truncation

does not impact on the lower bound calculations in this case.

Not unexpectedly, the LP data fits gave near-perfect matching to the $Y$ data sets produced by the gamma distribution convolutions. The $\mu$ values were then minimized subject to the constraint of maintaining different minimum degrees of $Y$ data fit. The resulting plots are shown in Fig. 4. For ease of comparison the goodness of fit is shown by correlation

coefficient $r$ values rather than the corresponding $\bar{D}_*$ values, recognising that $r$ is not a mathematical function of $\bar{D}_*$.

As noted earlier, defining a minimum-acceptable measure of fit is something of a judgement call. If $r = 0.9$ is employed for this role then it can be seen from Fig. 4 that this gives lower bounds which are both about 2/3 of the respective $\mu_T$ (lower bounds of 4 and 8 months respectively), presumably largely reflecting the gamma scale differences. Scatter plots indicating

the degree of model fits associated with $r = 0.9$ are shown in Fig. 5, together with the $P(\tau)$ probabilities which gave the minimized values of the respective $\mu$. In both cases the $P(\tau)$ probabilities which give the minimum values of mean transit times have no resemblance to the respective gamma transit time distributions, with the $P(\tau)$ probabilities for the gamma $\mu_T = 6$ months being particularly irregular.

It is interesting to note that even though the $P(\tau)$ obtained from data fitting for $\mu_T = 12$ months was upper-truncated, the choice of $N = 23$ is sufficiently large not to have effect when  minimising $\mu$. It is not essential therefore for $N$ to be sufficiently large to include the upper tails of transit time distributions in order to obtain lower bounds to $\mu_T$.

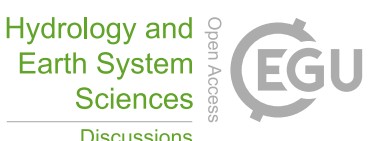

One obvious point of difference in Fig. 5 between the two $P(\tau)$ configurations is the higher value of $P(0)$ for the case of the gamma $\mu_T = 6$ months, which still maintained the $r = 0.9$ fit to data. This is something of an illusion, however, because constraining the $P(\tau)$ configuration here to avoid multiple models has minimal effect.

## 5  6 Discussion

It happened from the way that the example data were simulated that the best data fits gave lower bounds close to $\mu_T$. However, this will not be true in general and $\mu_*$ should not necessarily be interpreted as an approximation to $\mu_T$. This is because $\mu_*$ values are conditional on the tracer input concerned. For example, $\mu_*$ is never likely to be greater than about 4 years when $X$ data is dominated by seasonal variation of stable isotopes, but $\mu_T$ may be much greater (Stewart et al., 2010). It
could be argued in fact that all water ages should be reported as lower bounds unless the possibility of older water influencing $\mu_T$ can be discounted.

Application of the lower bounds could be in different contexts. For seasonal isotope $X$ inputs, plots of $\mu_*$ against goodness of fit from different catchments could serve for comparison of water throughput behaviours for time ranges of a few months to
a few years. Alternatively, tritium inputs might be used in some catchments so that $\mu_*$ gives a closer approximation to $\mu_T$, making allowance for tritium decay. However, as noted by Stewart et al., (2010), [3]H ages are currently ambiguous in many locations so $\mu_*$ might yield lower bounds considerably less than $\mu_T$.

One aspect not considered here is the error which might arise by applying linear models to catchment flow systems which
are in reality nonlinear. Kirchner (2016b) suggested that the young water fraction is robust against nonlinear effects when considering seasonal amplitude differences. It may happen that $\mu_*$ values are similarly robust but this would require further investigation.

If it happens that a catchment discharge is dominated by young water then it might be useful to select a small time resolution
for the $Y$ measurements. Stockinger et al. (2016) note that $\mu_T$ estimates may be influenced by data time resolutions and this will apply to $\mu_*$ also, with the possibility of $\mu_*$ values being artificially high if the $Y$ data time resolution is too coarse.

From the viewpoint of computation requirements, LP algorithms are typically efficient and can handle large numbers of linear variables subject to linear constraints. The illustrative applications here was easily set up in Excel spreadsheets and
computation was only a few seconds using a commercial LP utility for Excel. Computation times might be significant, however, if many binary variables were also included to prevent multiple modes in a set of time-varying $P(\tau)$ distributions.





## 7 Conclusion

Nonparametric lower bounds for mean transit times have potential to be of value in catchment studies and avoid some of the issues which arise when seeking direct estimation of $\mu_T$. However, confirmation of utility must come from application to real data. A necessary aspect of the lower bound approach for practical application would be for the bounds to show sufficient variation over different catchments types to provide useful insights. It would be particularly interesting if some studies could be initiated to obtain nonparametric lower bounds for selected data sets where age inferences have already been made by utilising parametric transit time distributions. It is hoped that this short communication will encourage such analyses and comparisons.

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

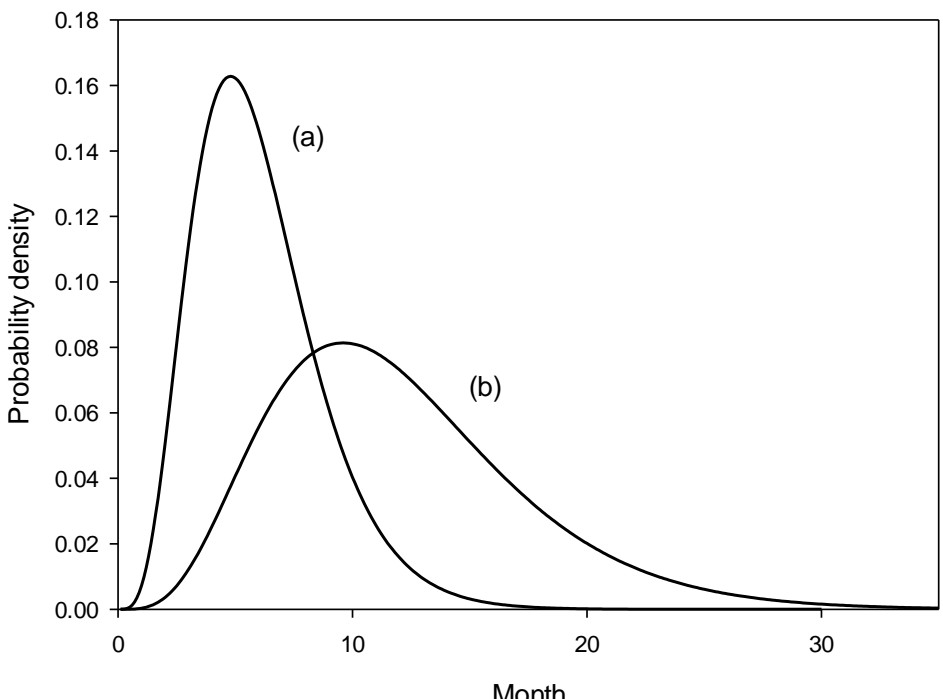

**Figure 1: The two gamma probability density functions utilised as transit time distributions. $\mu_T$ = 6 and 12 months for (a) and (b), respectively.**



**Figure 2: Simulated monthly time series of catchment tracer input and output. The blue plots, common to (a) and (b), show simulated tracer input with an annual cycle and random noise. Red plots show the transformation of the tracer input time series for a steady state linear catchment given convolution of the gamma transit time for $\mu_T$ = 6 months (a) and $\mu_T$ = 12 months (b).**





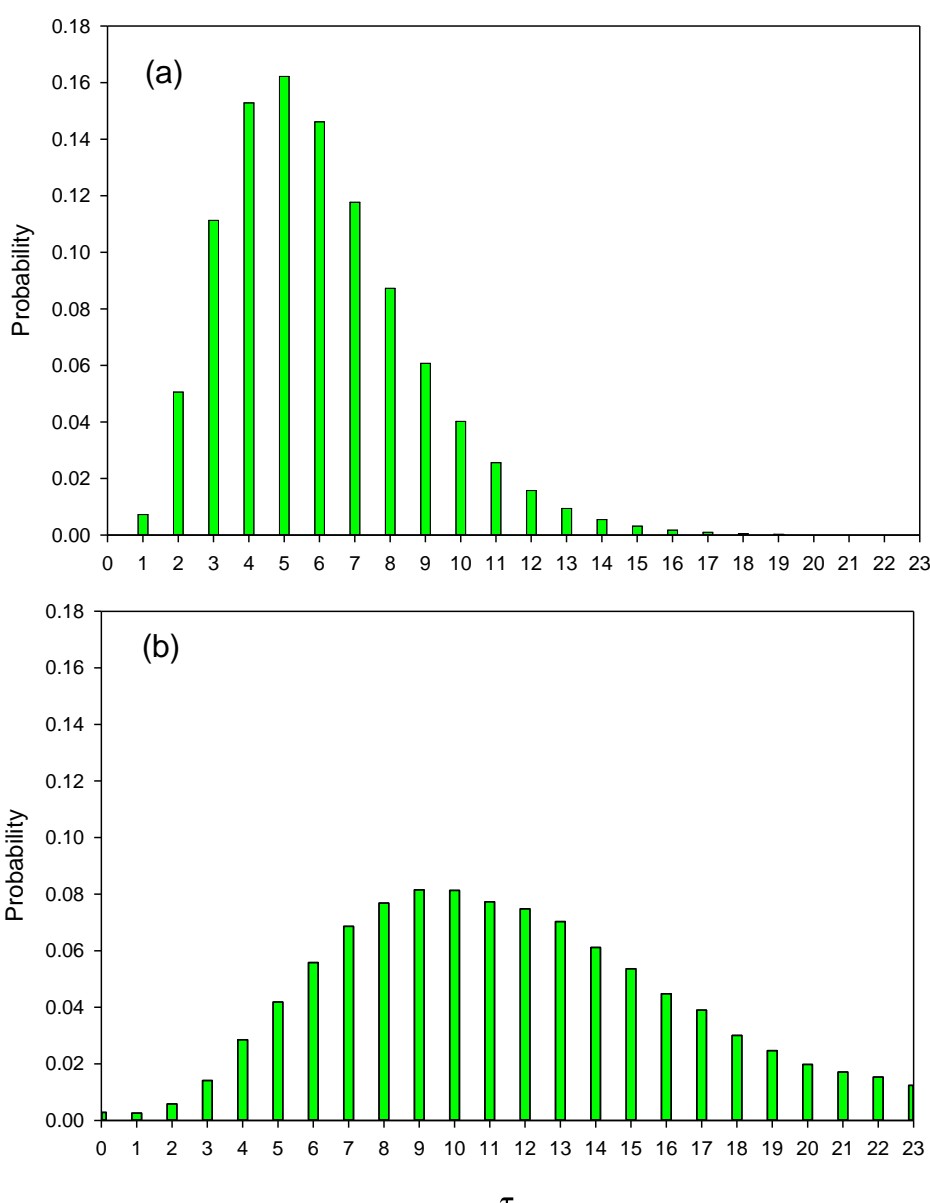

**Figure 3:** The $P(\tau)$ probabilities from best fit to the $Y$ data set for $\mu_T = 6$ months (a) and $\mu_T = 12$ months (b).





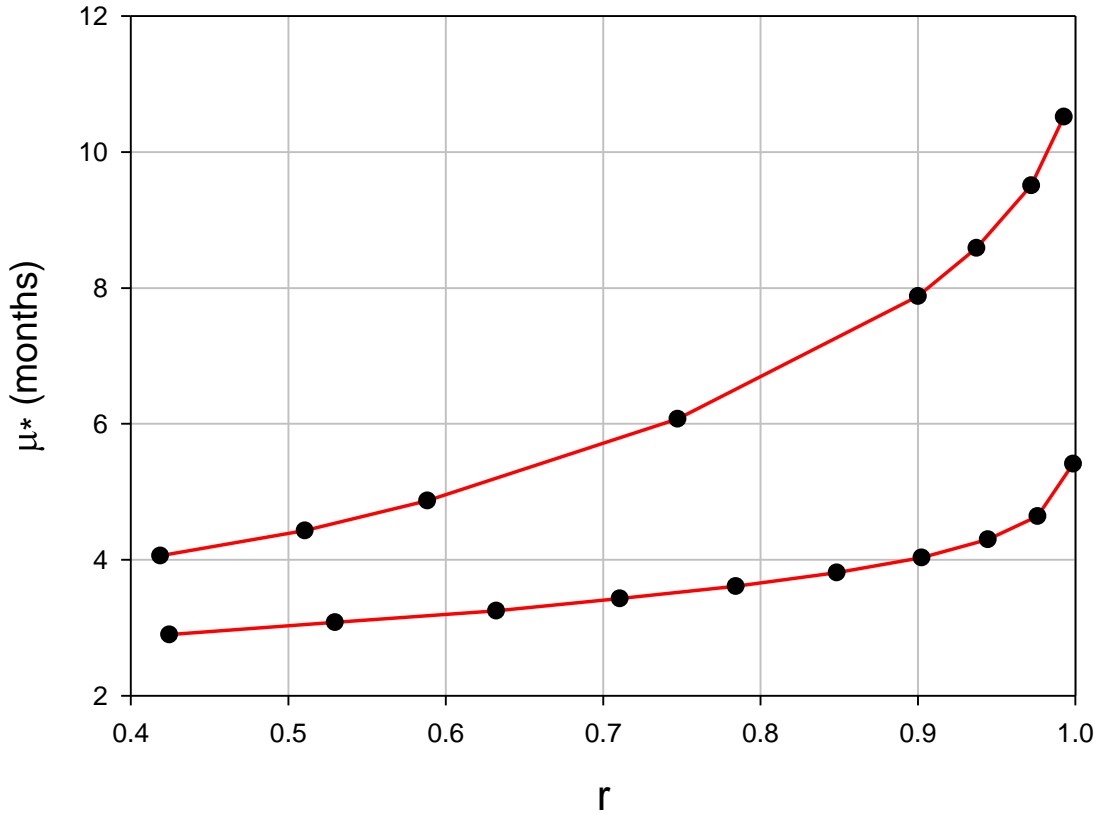

**Figure 4: Lower bound μ∗ plotted against minimum permitted goodness of fit (shown here as the correlation coefficient _r_) for μ_T = 6 months (lower plot) and μ_T = 12 months (upper plot).**







**Figure 5: (a), (b) Scatter plots of observed and model values from mean value minimizations with goodness of fit constrained to be not worse than an equivalent *r* value of 0.9. (c), (d) Corresponding $P(\tau)$ probabilities which created the minimized mean values $\mu_*$ for $\mu_T$ = 6 months and $\mu_T$ = 12 months, respectively.**