# Peer review of "Nonparametric lower bounds to mean transit times"

_Hydrology and Earth System Sciences, 2017_

## Referee Comment (RC1) · Anonymous Referee #1 · 24 Apr 2017

This paper proposes using a simple nonparametric linear programing approach to improve the actual used methods to obtain mean transit time, calculating instead the lower bounds of the mean transit time. The subject is of big relevance in hydrology, and coming up with newer and improved methods than the existing ones is always good for the scientific community. The paper was very well written However, there are some points I would like to address on this review. Finally, I agree with the author that it would be very interesting to use this approach in catchments with very well studied datasets with known mean transit times for a certain period of time. Note: This review does not analyze in depth the mathematical part of the approach because it goes beyond my expertise.

General comments:

1.) I would like to clarify to the author that mean transit time and mean residence time are not the same. The first one could be defined as the time elapsed of the water from

input to output, while the second one is the mean residence time of all the water you have in the system at a certain point. It is true that in the literature it has been used interchangeably in some occasions but fortunately that has happened less on the last 10 years if I am not wrong. I suggest three references where they explicitly point out the difference between both concepts; I think this could help clarify the idea. I don't think this affects the outcome of the approach the author is proposing, but I do think is very important to have these concepts very clear when working with them and specially when developing new methods involving them.

- McGuire, K., and McDonnell, J. (2006). A review and evaluation of catchment transit time modeling. JoH. http://doi.org/10.1016/j.jhydrol.2006.04.020

- Botter, G. et al. (2011). Catchment residence and travel time distributions: The master equation. GRL. http://doi.org/10.1029/2011GL047666

- Rinaldo, A., et al. (2015) Storage selection functions: A coherent framework for quantifying how catchments store and release water and solutes. WRR. http://doi.org/10.1002/2015WR017273

2.) Following the previous point, it grows the concern if the author has read enough of the available literature and methods proposed by other authors trying to obtain as well better transit time estimations. With this I am not intending on attacking the author in any way, but I am just concerned that if our interest is to advance in science we should take the time to observe what are the more appreciated methods proposed on the scientific community before we add one more to the pile. Doing this can help the author as well in improving his own approach. For example, in many of the transit time studies, a problem that limits having good transit time estimations is the storage (groundwater, evaporation, etc.) interaction with the system, for which there are several methods trying to attack this approach. Would the author be able to use this approach to consider as well the water that was in the system previous to the input studied?

3.) Here I copy some of the different studies that take into account the previous water

in the system in one way or another in addition to Botter et al (2011) and Rinaldo et al (2015) above mentioned:

- Heidbüchel, I. et al. (2012). The master transit time distribution of variable flow systems. Water Resources Research. http://doi.org/10.1029/2011WR011293

- Harman, C. J. (2015). Time-variable transit time distributions and transport: Theory and application to storage-dependent transport of chloride in a watershed. WRR. http://doi.org/10.1002/2014WR015707

-van der Velde, Y. et al. (2015). Consequences of mixing assumptions for time-variable travel time distributions. HP. http://doi.org/10.1002/hyp.10372

-Rinaldo, A. et al. (2016). Reply to comment by Porporato and Calabrese on "'Storage selection functions: A coherent framework for quantifying how catchments store and release water and solutes'". WRR http://doi.org/10.1002/2014WR015716

4.) As a suggestion for better understanding that the author might take or not. I think that some parameters could have more user friendly names. For example $\mu^*$ could have an L or low as sub-index. That would force to change the name of D to a sub-index like 'u', 'upp' or 'h' from highest. These small changes could make smoother for the reader to follow up the terms coming up throughout the paper.

5.) I am positively impressed that with this approach there is no need of catching the tail while using a gamma distribution. That is an upgrade for the gamma distribution methods. But it would be interesting as well to test different N values to see if it would provide different $\mu^*$ values. Meaning, is $\mu^*$ dependent on the size of N chosen?

6.) Would it be possible for unknown catchments to know that a $\mu^*$ with r=0.9 is 2/3 of the mean transit time? Or was this just a casual coincidence?

Specific comments:

7.) Page 2 Block 25: what are the X values with negative sub-indexes physically? Am

[Figure]

I correct if I assume them to be the previous inputs to the input I am studying? If that would be the case, then it would be other precipitation as well? Or is it stream water, or groundwater? Let's assume the case where my X values with positive sub-index are precipitation values, would that mean that all my X values with negative sub-indexes are precipitation values from days prior to the positive ones?

8.) P4 Eq 1: if we assume t=0 you would obtain an X-N which is not defined before.

9.) P6 B15 eq 5: Same as before, with t =$\epsilon$ there would be an X$\epsilon$-N not defined, that without saying that none of all the X obtained in that equations are defined due to the epsilon. I understand the author takes them as very small, but the nomenclature is confusing since those are sub-indexes. Perhaps it would be better if it's properly stated that for this case we will assume that the indexes with $\epsilon$ are as if there was no $\epsilon$.

10.) P6 B25: the progression of this sum is a bit confusing, I think it would go from $\omega$K-$\tau$+1, $\omega$K-$\tau$+2,... to $\omega$ K.

11.) P8 B25: I think the author meant $\beta$, instead of $\alpha$ in the text "The gamma scale parameter '$\beta$' is specified..."

12.) P10 B0: Could the author explain better how is it that higher value on P(0) an illusion? If I were to not consider that value there would be a shift on the $\mu$. I could not follow the author's idea.

13.) Fig 3: X axis is missing units (months I assume)

14.) It would be nicer if Fig 1, Fig2 and Fig 4 are done for black white printing and as well for color blind readers.

---

## Author Comment (AC1) · 28 Apr 2017

Reply to Reviewer # 1

My thanks taking the time for the review. Reviewer comments are in italic and responses follow.

*This paper proposes using a simple nonparametric linear programing approach to improve the actual used methods to obtain mean transit time, calculating instead the lower bounds of the mean transit time.*

There is no suggestion of offering an improvement over any previous methods that have been used to obtain mean transit time. The paper is concerned only with an approach to estimating a lower bound to mean transit time. The lower bound in some cases might be orders of magnitude less than the true mean transit time.

*I would like to clarify to the author that mean transit time and mean residence time are not the same.*

I guess we would all agree that the mean residence time for a hotel is the average of between when guests check in and when they check out. A quick overview of the literature reveals that this definition of mean residence time presently extends over many different fields. However, mean transit time serves perfectly well for the purposes of the paper so the very few mentions of mean residence time will be removed.

*Following the previous point, it grows the concern if the author has read enough of the available literature and methods proposed by other authors trying to obtain as well better transit time estimations.*

As noted earlier, this technical note is not concerned with seeking better transit time estimation, but rather with proposing a model-independent nonparametric lower bound to mean transit time. That is, the aim is not to find where the mean is, but to define a region where it is not. There is certainly a proliferation of papers relating to transit time concepts, hydrological models leading to various transit time distributions, as well as papers relating to *ad hoc* transit time distributions like the gamma distribution. This extensive literature is not cited because it is unrelated directly to the specific issue of nonparametric lower bound estimation of mean transit time. Perhaps the best approach would be in the Introduction to direct readers to, for example, the cited references in Kirchner 2016a&b . A search for methods directly connected to lower bound estimation for mean transit times did not turn up anything in the hydrological literature. I would of course welcome notification of any such references in journals or texts from hydrology or other fields.

*Would the author be able to use this approach to consider as well the water that was in the system previous to the input studied?*

The analysis in the paper is not concerned with water as such but with some form of tracer particle passing into the system, observable as a given tracer input time series. A requirement of the method is that the tracer input data extends far enough back in time so that all tracer particles in the system can be mapped back in principle to the recorded tracer input time series. The method would not be applicable with significant amounts of tracer particles present that could not be related to the recorded tracer input time series.

*As a suggestion for better understanding that the author might take or not. I think that some parameters could have more user friendly names. For example μ\* could have an L or low as sub-index. That would force to change the name of D to a subindex like 'u', 'upp' or 'h' from highest. These small changes could make smoother for the reader to follow up the terms coming up throughout the paper."*

My thanks for these suggestions – they will be incorporated in the event that the paper is advanced to accepted status.

*I am positively impressed that with this approach there is no need of catching the tail while using a gamma distribution. That is an upgrade for the gamma distribution methods.*

Many thanks for the support here. However, I am a little confused as to the meaning. A specific gamma distribution was used as an entirely arbitrary choice of a travel time distribution, but just for the purposes of simulating example data with a known mean transit time. However, the method itself is non-parametric and not in any way connected to the gamma distribution or any other parametric transit time distribution.

*But it would be interesting as well to test different N values to see if it would provide different μ\* values. Meaning, is μ\* dependent on the size of N chosen?*

As noted in the paper, the choice of $N$ is not critical as long as it is sufficiently large not to have influence later in the minimization when seeking a lower bound for $\mu_T$. In the LP minimising operation the probability distribution shifts to the left to minimize the distribution mean value, subject to the constraints. When the distribution thus created has probabilities that are zero in the upper range of the distribution, this confirms that increasing $N$ will not affect the minimization outcome. This can be seen, for example, in Fig. 5 c & d.

*Would it be possible for unknown catchments to know that a μ\* with r=0.9 is 2/3 of the mean transit time? Or was this just a casual coincidence?*

This is with respect to Fig. 4. Sadly, it is just a coincidence. Each real-world catchment will have its own characteristics and data quality can also influence $r$.

*Page 2 Block 25: what are the X values with negative sub-indexes physically? Am I correct if I assume them to be the previous inputs to the input I am studying? If that would be the case, then it would be other precipitation as well? Or is it stream water, or groundwater? Let's assume the case where my X values with positive sub-index are precipitation values, would that mean that all my X values with negative sub-indexes are precipitation values from days prior to the positive ones?*

The negative subscripts for X indicate the recorded time series of tracer input prior to the time of the first Y output variable. This prior period of off-lap is required because the first Y value ($t =1$) will be influenced by all the prior X values as far back as is determined by the transit time distribution. It seems convenient to have negative subscripts for the X values to emphasize the off-lap. The physical interpretation of the X values depends on the situation – they could, for example, be flux-weighted $^{18}O$ values obtained from a rainfall time series for the catchment. So, yes, the X values with negative indices are from prior to the positive ones.

*P4 Eq 1: if we assume t=0 you would obtain an X-N which is not defined before*

There is only a requirement for seeking to match the available set of Y values, that is $Y_1$ , $Y_2$, …$Y_K$ and there is no $Y_0$ . That is, $t$ in Eq 1 can be any of $t = 1, 2, …$ K but there is no $t = 0$ involved in Eq.1. My thanks for pointing out the lack of clarity here. The potential for confusion will be avoided by having " $1 \leq t \leq K$ " to the right of Eq. (1).

*P6 B15 eq 5: Same as before, with $t = \varepsilon$ there would be an $X_{\varepsilon -N}$ not defined, that without saying that none of all the X obtained in that equations are defined due to the epsilon. I understand the author takes them as very small, but the nomenclature is confusing since those are sub-indexes. Perhaps it would be better if it's properly stated that for this case we will assume that the indexes with $\varepsilon$ are as if there was no $\varepsilon$.*

Again, the issue has arisen because Eq. (5) should have had $1 \leq t \leq K$ inserted to the right. My apologies for this omission.

The nomenclature has been reworked somewhat (see suggested new text following the **** on the next page).

*P6 B25: the progression of this sum is a bit confusing, I think it would go from*
$\omega_{K-\tau+1}, \omega_{K-\tau+2},: : :$ *to* $\omega_K$.

This portion of the paper could certainly have been written more clearly. My suggestion for a re-write of both text and equations for this part of the paper are as below (there would also need to be a corresponding modification to Eq. (1) for consistency). Hopefully this new text and symbolism will read better:

\*\*\*\*\*\*\*\*\*\*\*\*\*\*\*\*\*\*\*\*\*\*\*

Because the different $P(\tau)$ distributions have different time origins, a given discrete distribution is now symbolised as $P_i(\tau)$, where $0 \leq \tau \leq N$ as before. That is, $P_i(\tau)$ is the transit time distribution for tracer particles which arrive in the catchment at time $i$, where $1\text{-N} \leq i \leq K$. At a given time $t$ the model-predicted tracer output $\hat{Y}_t$ at the recording site is given by:

$$\hat{Y}_t = \theta \sum_{\tau=0}^{N} X_{t-\tau} P_{t-\tau}(\tau) = \sum_{\tau=0}^{N} X_{t-\tau} \omega_{t-\tau}(\tau) \qquad 1 \leq t \leq K \qquad (5)$$

The data fit expression giving the least mean absolute deviation from the *Y* data is therefore found from the minimization operation:

$$\text{MINIMIZE} \quad K^{-1} \sum_{t=1}^{K} \left| Y_t - \sum_{\tau=0}^{N} X_{t-\tau} \omega_{t-\tau}(\tau) \right| \qquad (6)$$

subject to the equality constraints:

$$\sum_{\tau=0}^{N} \omega_{1-N}(\tau) = \theta \qquad , \qquad \sum_{\tau=0}^{N} \omega_{2-N}(\tau) = \theta \qquad , \qquad \dots \sum_{\tau=0}^{N} \omega_0(\tau) = \theta$$

$$\sum_{\tau=0}^{N} \omega_1(\tau) = \theta \quad , \quad \sum_{\tau=0}^{N} \omega_2(\tau) = \theta \quad , \quad \dots \sum_{\tau=0}^{N} \omega_K(\tau) = \theta$$

where the scale parameter $\theta$ is obtained from the minimization, which consists of finding numerical values for each $\omega_i(\tau)$ such that Eq. (6) is minimised. The individual $P_i(\tau)$ probabilities are then found by rescaling with $\theta$ as before. If an acceptable data fit cannot be achieved at this point then a more linear flexible model could be evaluated, allowing some degree of variation of the respective $\mu$ values.

\*\*\*\*\*\*\*\*\*\*\*\*\*\*\*\*\*\*\*\*

*P8 B25: I think the author meant β, instead of α in the text "The gamma scale parameter 'β' is specified: : :"*

Yes – my thanks for picking the use of the wrong symbol.

*P10 B0: Could the author explain better how is it that higher value on P(0) an illusion? If I were to not consider that value there would be a shift on the μ. I could not follow the author's idea.*

What was meant here is that having a mode at P(0) is not a sensitive factor for determining the calculated value of the lower bound. For example, if additional constraints are imposed so there cannot be a mode at P(0) from the minimisation process, there is in fact minimal change to the value of the calculated lower bound in this instance. This is a good illustration for showing that the $P(\tau)$ values arising from the minimisation should be regarded as uninformative. Some text making the clarification will be added

*Fig 3: X axis is missing units (months I assume)*

Yes .. there are units missing as noted. The horizontal axis needs to read " $\tau$ (months) "

*It would be nicer if Fig 1, Fig2 and Fig 4 are done for black white printing and as well for color blind readers.*

Yes – the colour figures were for review purposes.

---

## Referee Comment (RC2) · Anonymous Referee #2 · 8 May 2017

GENERAL COMMENTS This technical note proposes a method to estimate lower
bounds to mean transit time in hydrological systems. The author makes use of non-
parametric transit time distributions (TTDs), obtained by fitting tracer data in precip-
itation and streamflow. In particular, a new index is proposed which represents the
minimum value of mean transit time that is compatible with a user-defined goodness of
fit of the tracer data. The note is well organized and clearly written, but I have some ma-
jor concern on the significance of the results. While I believe the use of non-parametric
TTDs is an interesting and under-explored topic, the index proposed by the author
does not, in my opinion, provide improved understanding of hydrological processes. I
summarize below my major points:

1) What is the usefulness of a (potentially arbitrary) lower bound to mean transit times?
The motivations behind this new index are in my opinion not strong enough. The author
mentions (page 2, line 15-19) the connection with long history of mean TTD application

in hydrological studies, but it is not clear what the connection could actually be. Also note that the transit time literature has notably evolved in the last years (e.g. Kirchner 2016a,b, already cited by the author, but also the time-variant approaches to TTD modeling described by Rinaldo et al., (2015)), with the concept of stationary mean TTD now becoming rather obsolete.

2) As noted by the author (page 3 line 10), the use of nonparametric TTDs typically leads to several different distributions (with different means) that provide equally good results. Hence, there exists a whole distribution of mean transit times that allow fitting tracer data at the user-specified goodness of fit. What is not clear to me is why the author focuses on the minimum value, which is not a robust statistic, instead of focusing on other properties of this distribution which could better highlight the difficult determination of mean transit times;

3) The time-varying example is restricted to the very specific case of TTDs with different shapes but equal mean, which is a strong limit. Also, the time-variance of the distributions seems to be just an additional degree of freedom in the minimization process, which it is not related to any physical process that may lead TTDs to change with time. Although this is technically a form of time-variance, it is not the one that is usually pursued in catchment studies;

4) A more convincing proof-of-concept application should be provided. The synthetic dataset used by the author is generated through a TTD which is not realistic for most watershed (the parameter alpha is typically < 1, see Godsey et al., 2010). In my opinion, to convince the reader of the actual usefulness of this new index, a real-data example should be provided, although this option would require converting the technical note into a regular article;

MINOR COMMENTS

Page 2, lines 5-7: there are some interesting (although simplified) examples in the literature where the shape of the probability distribution is derived theoretically. I would

suggest referring to Kirchner et al., (2001) and Leray et al., (2016).

Equation (1) and p.4, l.5: it should be stated explicitly why theta can be different from 1, as according to the hypothesis of ideal tracer it should always be equal to 1.

P.3, l.6-8: I did not understand this sentence.

P.5, l.21-23: this sound very speculative and it is unclear what could actually be compared from one catchment to the next.

Section 4: the "time-variant" case is actually a very particular case of time-variance. This should be clearly specified everywhere (e.g. in the title of section 4)

SUGGESTED LITERATURE

Godsey, S. E., Aas, W., Clair, T. a., de Wit, H. a., Fernandez, I. J., Kahl, J. S., . . . Kirchner, J. W. (2010). Generality of fractal 1/f scaling in catchment tracer time series, and its implications for catchment travel time distributions. Hydrological Processes, 24(12), 1660–1671. http://doi.org/10.1002/hyp.7677

Kirchner, J. W., Feng, X., & Neal, C. (2001). Catchment-scale advection and dispersion as a mechanism for fractal scaling in stream tracer concentrations. Journal of Hydrology, 254(1–4), 82–101. http://doi.org/10.1016/S0022-1694(01)00487-5

Leray, S., Engdahl, N.B., Massoudieh, A., Bresciani, E., and McCallum, J., (2016), Residence time distributions for hydrologic systems: Mechanistic foundations and steady-state analytical solutions, Journal of Hydrology, 543, 67-87, https://doi.org/10.1016/j.jhydrol.2016.01.068.

Rinaldo, A., Benettin, P., Harman, C. J., Hrachowitz, M., McGuire, K. J., van der Velde, Y., . . . Botter, G. (2015). Storage selection functions: A coherent framework for quantifying how catchments store and release water and solutes. Water Resources Research, 51(6), 4840–4847. http://doi.org/10.1002/2015WR017273

---

## Author Comment (AC2) · 11 May 2017

My thanks to reviewer #2 for the comments offered. Comments are copied below in italics and responses follow.

*The index proposed by the author does not, in my opinion, provide improved understanding of hydrological processes.*

There is no implication that the paper seeks to improve understanding of hydrological processes. All a lower bound can do is to give an indication that the true mean transit time is probably greater than some minimum value. That's not about hydrological processes but it does give some measure of catchment hydrological information and has value to that extent – provided the bound is sufficiently removed from zero of course.

*1) What is the usefulness of a (potentially arbitrary) lower bound to mean transit times? The motivations behind this new index are in my opinion not strong enough. The author mentions (page 2, line 15-19) the connection with long history of mean TTD application in hydrological studies, but it is not clear what the connection could actually be. Also note that the transit time literature has notably evolved in the last years (e.g. Kirchner 2016a,b, already cited by the author, but also the time-variant approaches to TTD modeling described by Rinaldo et al., (2015)), with the concept of stationary mean TTD now becoming rather obsolete.*

As noted above, the idea of a lower bound to mean transit time can provide some information on a catchment system, to the extent of saying where the mean transit time is probably not. Recognizing at the same time that there is an element of subjectivity about the bound, because specification is required as to what a "good" fit is.
With respect to "connection" the wording should be clarified. It was just meant that the mean transit time still survives here as being referenced by the lower bound – nothing more than that. The contrast was with Kirchner's 2016a index of fraction of young water, which does not mention mean transit time.
There is no disagreement that the idea that the stationary transit time distribution concept may be tending toward obsolescence. However, in constructing the paper it was natural to first consider the well-known special case of stationary transit time distributions.

*2) As noted by the author (page 3 line 10), the use of nonparametric TTDs typically leads to several different distributions (with different means) that provide equally good results. Hence, there exists a whole distribution of mean transit times that allow fitting tracer data at the user-specified goodness of fit. What is not clear to me is why the author focuses on the minimum value, which is not a robust statistic, instead of focusing on other properties of this distribution which could better highlight the difficult determination of mean transit times.*

Not just leads to several different distributions, but leads to an infinity of different distributions, some of which may share common means. This in fact is why there is focus on the lower bound. Any number of transit time distributions with different distribution forms and different means could be compatible with the data to a given level of fit. That is, there is no information that can be extracted about the transit time distribution or mean transit time. However, the more useful approach is to ask the question "how small can the mean transit time be (whatever the transit time distribution might be) before there is unacceptable departure from the data?" This defines the lower bound, which does provide some information provided it is sufficiently removed from zero.
The paper is not concerned with a minimum value in the sense of a sample minimum value and the proposed bound should not be flattered by calling it a "statistic", robust or otherwise. The bound is essentially a subjective indicator of where the mean is probably not, using "probably" in a non-mathematical sense. This is not to say that we should abandon seeking to estimate catchment transit time distributions, it's just that the nonparametric approach is not likely to be helpful for this end. However, it might be possible to use a similar approach to ask "what is the largest and smallest proportion of young tracer that any transit time distribution could tolerate before data mismatch sets in?" This could be a topic for further investigation but is beyond the scope of this brief technical note.

*3) The time-varying example is restricted to the very specific case of TTDs with different shapes but equal mean, which is a strong limit. Also, the time-variance of the distributions seems to be just an additional degree of freedom in the minimization process, which it is not related to any physical process that may lead TTDs to change with time. Although this is technically a form of time-variance, it is not the one that is usually pursued in catchment studies.*

Yes, there is no question that an example model of a sequence of transit time distributions with common means but varying forms is an overly-constrained representation of time variability. It would be best to substitute with an alternative example where the means vary as well as the distribution forms. However, as noted in the last paragraph of Section 4, there still needs to be some real-world constraints added to provide an upper limit to the extent to which each transit time mean may differ from the previous one. In a given application further constraints might be added by a user to force consistency with some known physical process, to the extent that such information is available. This could lead to a significant LP model with many variables and constraints. However, LP methodology is now well developed and efficient. For the purposes of illustration in a revised paper, the constraints will just be limiting the degree to which the transit time means may differ from each other, while otherwise allowing arbitrary freedom of distribution forms. This gives the maximum flexibility and therefore the lowest possible lower bound subject to limiting the variability of the means.

*4) A more convincing proof-of-concept application should be provided. The synthetic dataset used by the author is generated through a TTD which is not realistic for most watershed (the parameter alpha is typically < 1, see Godsey et al., 2010). In my opinion, to convince the reader of the actual usefulness of this new index, a real-data example should be provided, although this option would require converting the technical note into a regular article.*

The reference to the gamma shape parameter here is something of a distraction, because there is implication that the gamma distribution has theoretical justification over other distributions for general application to catchment transit time distributions. It does not – and neither does any other parametric distribution. The cited reference shows that the gamma distribution fits data better than its special case of the exponential distribution, which is not surprising when a one-parameter distribution has to compete with a two-parameter alternative to match data. The exponential distribution is also a special case of the Weibull distribution and an equivalent analysis could have been carried out which would have rejected the one-parameter exponential distribution in favour of a two-parameter Weibull alternative. But this would not imply that the Weibull distribution has theoretical justification.

The reviewer point being made therefore is not actually with respect to a gamma distribution shape parameter value, but rather pointing out that it would be useful to have data simulated from a heavy-tailed distribution as such forms are often consistent with observations. The gamma distribution is no less arbitrary than any other for defining a transit time distribution for simulation purposes, so there is no problem with an additional example with a gamma shape parameter <1. The obvious comparison would be to keep the same mean travel time as the original example and see what impact the change in distribution shape has on the lower bound. If it happens that the lower bound in such instances is forced toward zero then the lower bound concept would certainly be restricted in its application, so it is a good review suggestion.

There is no suggestion of attempting in a short communication to convince a reader of the usefulness of the lower bound. Even a single application to real data would not be particularly convincing. The aim is simply to raise curiosity sufficiently for others to try it out and see how it goes. The method is easy to apply and perhaps some example spreadsheets could be included in a revised version to speed the process of use by others. Such application to real data is deliberately avoided here because (i) it is good to illustrate the lower bound with known mean transit times, which are always unknown in reality, (ii) owners of real data will inevitably have some feel for the catchments involved and are in the best position to judge whether the lower bound is useful to them or not. Preference therefore is to maintain the paper in a technical note status.

MINOR COMMENTS

*Page 2, lines 5-7: there are some interesting (although simplified) examples in the literature where the shape of the probability distribution is derived theoretically. I would suggest referring to Kirchner et al., (2001) and Leray et al., (2016).*

The manuscript text here is:

"Generally speaking, no probability distribution can claim particular theoretical justification in any form of hydrological study unless the situation of the physical environment matches the statistical characterization of the distribution concerned. Such characterization of course excludes fortuitous empirical matching of a given distribution to recorded or simulated data."

In fact there will never be transit time distributions which have mathematical derivation with respect to real-world catchment systems. As noted by Kirchner (2016a), application of any derived distribution to real catchments is simply a "hope" that it might be the right distribution. The literature transit time distributions are essentially irrelevant because they have been derived not for hydrological systems, but for models of hydrological systems. The Leray et al. (2016) paper in fact would have been more correctly titled "Residence time distributions for models of hydrologic systems …" . However, there is certainly value in including references to the two papers concerned as examples of model-derived transit time distributions. The references will be included in any revised version.

*Equation (1) and p.4, l.5: it should be stated explicitly why theta can be different from 1, as according to the hypothesis of ideal tracer it should always be equal to 1.*

Yes – some added text would be helpful.

*P.3, l.6-8: I did not understand this sentence.*

The text concerned is:

"The inclusion of $\tau = 0$ may seem unusual because it implies some tracer being instantaneously transported to the recording site. A finite probability of zero time is of practical value, however, because it ensures that any calculated lower bound to $\mu_T$ is not slightly higher than need be, as would be the case if all transit times were bounded below at 1.0."

This issue arises because the transit time distribution utilised is a discrete distribution defined over the integers. Assigning a zero probability to $\tau = 0$ would have the effect of slightly increasing the mean of the distribution. On the other hand, permitting a non-zero probability for $\tau = 0$ implies an element of instantaneous tracer movement to the recording site. Faced with these two options, the nonzero probability of $\tau = 0$ was selected because the effect will be to make the lower bound as small as possible. This is all essentially a rounding effect resulting from approximating a continuous distribution with a discrete distribution. Some text along these lines will be added to any revised version.

*P.5, l.21-23: this sounds very speculative and it is unclear what could actually be compared from one catchment to the next.*

The text concerned is:

" There is therefore a degree of subjectivity with the lower bound because perceptions may differ over what constitutes a just-acceptable fit. Nonetheless, plots of $\mu_*$ against $\bar{D}_*$ give useful summaries of the strength of evidence for a given lower bound, which in principle can be compared from one catchment to the next."

The idea here is just that lower bounds might be compared for different catchments. For example, if one catchment happened to have a much higher lower bound than another then that would indicate that there is some degree of confidence (using the word loosely) that the higher-bound catchment has a long mean residence time. On the other hand, there would be less certainty about what the mean transit time might be for the catchment with the lower bound. The text will be reworded in any revised version to make this clearer.

*Section 4: the "time-variant" case is actually a very particular case of time-variance.*
*This should be clearly specified everywhere (e.g. in the title of section 4)*

The current particular case of time-variance will be replaced with general time variance, with both time-varying distributions and time-varying means.

---

## Referee Comment (RC3) · Anonymous Referee #3 · 18 May 2017

The technical mote provides tools to derive numerically lower bonds to mean transit times using tracer time series. I'm not in favor of publication of this note for the reasons discussed below.

1. relevance. As noted by other reviewers, the practical and/or theoretical relevance of this paper is unclear to me. Getting lower bonds for Mean Transit Times does not seem to be an obviously relevant scientific problem; therefore, the author should better discuss the important implications of this work for hydrology and interpretation of environmental time-series. Interestingly, the paper by Jim Kirchner who apparently inspired this work, suggested that the mean transit time is a poor representation of this type of systems (as noted by the author in the introduction). In fact, tracer dynamics are mostly driven by the fraction of young water, and its time-variability.

2. tools. there are several papers that have shown unambiguously why the lumped convolution approach is misleading. It is quite surprising to see that the lumped convolution approach is still used (in this case in a discrete fashion). Therefore, in my view section 3 is completely useless - as that lumped formulation is not able to describe real-wold catchment dynamics. Note that Kirchner [2016a] used that approach only to explain the idea of aggregation bias in simple terms. Here the steady nature of TTDs (or MTTS) is a fundamental assumption (in fact, all the numerical examples shown in the paper refer to steady state TTDs).

3. tools. I have several problems also with the time varying version of the approach (section 4). first of all, the author apparently uses "forward distributions" and not "backword distributions" in equation (5). If this is correct, than the convolution in eq. (5) should refer to mass fluxes and not to concentrations, as in section 3 (see also the periodic nature of the input in Figure 2). Overall, the insisted use of the terminology "tracer time-series" is confusing. The meaning (and the nature) of the kernel linking input-output signals in environmental systems changes, depending on the quantity involved (concentrations, mass fluxes, water fluxes), as extensively discussed in the literature of the 70s by Niemi, Zuber and many others. Moreover, the constraint that the mean of the time-varying distributions is the same (Section 4) is untenable. There are tens of experimental and theoretical studies that show unambiguously how the mean travel time changes depending on hydrologic conditions (e.g. catchment wetness).

4. results. Based on my previous comment I find quite surprising that all the numerical examples refer to the steady state system.

5. as noted by other reviewers, the referencing is definitely inappropriate. I think the author is missing a huge number of papers about travel time formulation and application. As an example, the references to Kirchaner, Selle and Peralta-Tapia whan talking about TTD time-variance should be properly complemented by those works where the idea of time-variant TTDS has been proposed, proved theoretically and then applied. The relevant missing papers are too much to list them here.

Overall my impression is that this work is a nice mathematical exercise that unfortunately disregards the physical processes involved in catchment transport processes, and (as such) it as a reduced potential for real world hydrological applications.

---

## Referee Comment (RC4) · Anonymous Referee #4 · 20 May 2017

This technical note proposes that linear programming methods can be used to fit discrete nonparametric transit time distributions to tracer data, and a lower bound for the mean transit time can be determined among all such distributions that achieve an acceptable goodness of fit to the data.

The proposed method may be a useful contribution if a) its underlying assumptions can be shown to be correct (at least to a good enough approximation), or if, nevertheless, b) its results can be shown to be robust and reliable under realistic benchmark tests, which will include both realistic data errors and deviations of the real world conditions from the idealized assumptions of the method.

Unfortunately, condition (a) is not met, because the behavior of real-world catchments is widely recognized to be inconsistent with the linearity assumptions that underlie the method presented, and the illustrative example presented in section 5 falls far short of the requirements of condition (b). Specific comments on each section follow.

[Figure]

Section 1 (introduction):

The manuscript does not indicate much familiarity with the extensive literature on transit time estimation, or with the fundamentals of transit time models. To take just one example, the introduction confuses conditions that are SUFFICIENT to generate a given transit time distribution, with conditions that are NECESSARY to do so:

"In this regard it could be noted the widespread use of the well-mixed model (exponential distribution of transit times) seems particularly inappropriate because the assumption is that in a given small interval of time all tracer particles must have equal probability of passing out the catchment exit.... This must apply regardless of tracer particle location on the land surface or below ground... Similarly, gamma distributions do not warrant special consideration as transit-time distributions outside of idealised situations, except for the special case where the catchment largely comprises a cascade of well-mixed lakes (gamma distributions obtained as sums of independent exponential random variables)."

While it is true that equal probability of exit is SUFFICIENT to generate an exponential transit time distribution, it is not the case that this is the ONLY way that such a distribution can arise (one can imagine, hypothetically, a series of flowpaths whose lengths are exponentially distributed, or an aquifer whose permeability decreases systematically with depth. In both cases one could obtain an exponential distribution without equal probabilities of exit for every particle in the system.

The statement about gamma distributions is likewise misplaced. Nash cascades can indeed generate gamma distributions, but only for integer shape factors >=1. In contrast, the empirically determined shape factors among the 20 catchments analyzed by Godsey et al. (2010) ranged from about 0.3 to 0.8. None were even close to 1. Approximate gamma distributions with shape factors near 0.5, consistent with the Godsey et al. findings, can potentially arise from advection and dispersion with spatially distributed inputs (Kirchener et al. 2001).

Sections 2 and 3 (definitions and steady-state case):

From a purely theoretical standpoint I do not see any problems here, but I am not an expert in linear programming techniques. From a practical standpoint the steady-state case is relevant only as a (potentially rather poor) approximation to the real world.

Section 4 (non-steady-state case):

The time-varying case presented here, in which the shape of the transit time distribution can change but its mean must stay the same, is inconsistent with the entire literature on catchment nonstationarity. Even theoretically, it is very difficult to imagine any catchment that could possibly work this way. This section may be interesting from a mathematical standpoint but is completely irrelevant to the real world.

Section 5 (illustration):

There are very substantial problems here:

a) The gamma PDF's (with shape factor of 5) that are used to generate the test time series bear no resemblance to real-world catchment PDF's (with shape factor well below 1). There is no reason to assume that a method that works with such an unrealistic test time series will necessarily work with a more realistic one.

b) The nonparametric TTD is unrealistically truncated at 23 months. Of course, given that the generating distributions – see (a) above – have trivial tails beyond 23 months, the truncation effects are small. But in the real world, where one cannot know this in advance, what could be the justification for not extending the nonparametric TTD to much longer lags? In that case, of course, the computations would become more difficult and, more importantly, the solutions would become much less constrained.

c) The truncation of the nonparametric TTD automatically imposes bounds on the mean – between 0 and 23 months in theory, but in practice, with any dispersion (either physical or numerical), the range will be narrower and the tendency will be more toward the center. Thus it is guaranteed that the solution will not be that far from 6 or 12 months,

which are known in advance to be the correct answers.

d) The simulated time series are unrealistically long (not many sites have 21 years of tracer data).

e) The simulated time series are unrealistically PERFECT. There are no measurement errors in either the input or output time series. Random numbers are added to the square wave in order to create the X values, but then the Y values are convolved with zero error, and then the X and Y values are used – with zero error – in the proposed LP inversion technique. This bears no resemblance to the real-world inversion problem, where both the inputs and outputs will have errors. Furthermore, those errors are potentially catastrophic for an poorly constrained inversion technique like the proposed LP method.

f) In summary, is such a contrived test case that it gives no useful information about the robustness or the practical utility of the proposed method. There is no guarantee that, in a more realistic test case, the proposed method would give a meaningful lower bound estimator. Indeed, the result could potentially even exceed the true mean.

---

## Author Comment (AC3) · 23 May 2017

My thanks to Reviewer #4 for a detailed range of comments. The various reviewer comments are set out below in italic, responses follow in plain text.

*The manuscript does not indicate much familiarity with the extensive literature on transit*
*time estimation, or with the fundamentals of transit time models.*

There is certainly an extensive literature on transit time estimation and transit time models. The omission is deliberate.  Such analyses are invariably parametric and model-based, and review of that large literature would be peripheral to the nonparametric model-free theme of a short technical note. My personal view is that for brief communications of this type references should strictly focus only on the relevant, even though inevitably a number of reviewer publications may not be cited.

*The proposed method may be a useful contribution if a) its underlying assumptions can*
*be shown to be correct (at least to a good enough approximation), or if, nevertheless,*
*b) its results can be shown to be robust and reliable under realistic benchmark tests,*
*which will include both realistic data errors and deviations of the real world conditions*
*from the idealized assumptions of the method.*
*Unfortunately, condition (a) is not met, because the behavior of real-world catchments*
*is widely recognized to be inconsistent with the linearity assumptions that underlie the*
*method presented, and the illustrative example presented in section 5 falls far short of*
*the requirements of condition (b). Specific comments on each section follow.*

With respect to (a), real-world transit time distributions do always exist, regardless of their variability with time and the nature linear or nonlinear hydrological processes which give rise to them. In its most general form as a time sequence of arbitrary discrete transit time histograms, the nonparametric model can conceptually match any possible time sequence of transit time distributions within the range of the discrete histograms, while not explicitly modelling any nonlinear hydrological process. The lower bound should therefore still be applicable because any system of constraints incorporated to reflect nonlinear processes would have the effect of making the model less flexible and the lower bound would then be greater than when the minimisation process is applied to the unconstrained model.  In practice there will probably need to be some realistic constraints applied to time-varying nonparametric histograms such as restricting the degree to which the nonparametric distribution means are permitted to differ from one distribution to the next in sequence.

With respect to (b) there is no argument that a more realistic illustrative example is required – also noted also in responses to other reviewers. As was also noted in other responses, however, the aim of the brief technical note is not to carry out exhaustive evaluations to "prove" the applicability of the method. Rather, the intention is simply to use (inevitably somewhat idealised) examples to encourage individuals with local catchment knowledge to evaluate the lower bound approach when applied to their recorded data sets. Local knowledge is a critical factor because the greater the degree of constraints that can be placed on the histogram sequence, the higher the lower bound will be for a given degree of fit.

*While it is true that equal probability of exit is SUFFICIENT to generate an exponential*
*transit time distribution, it is not the case that this is the ONLY way that such a distribution can arise (one can imagine, hypothetically, a series of flowpaths whose lengths*
*are exponentially distributed, or an aquifer whose permeability decreases systematically with depth. In both cases one could obtain an exponential distribution without*
*equal probabilities of exit for every particle in the system.*

Equal probability of exit is in fact a necessary condition for a well-mixed water store to give an exponential transit time distribution. This is because the definition of a well-mixed water store is equal probability of exit.

The reference is the paper was clearly only to the lack of general applicability of the exponential distribution with respect to its justification in terms of well-mixed stores – which will only arise in rather restricted conditions. It is regretful however, that the explicit statement was not made in the paper that outside of the well-mixed model there

is also no theoretical justification in the real world for exponential transit time distributions – nor any other parametric transit time distribution for that matter.

Considering now the two supposed alternative exponential models proposed by the reviewer:

With respect to the flowpath model, the proposed series of flowpaths whose lengths are exponentially distributed is a special case of a stream tube model. It is indeed a *very* special case because is required that nature should for some reason create a situation such that a large number of stream tubes should (i) have an exponential distribution of lengths, (ii) have exactly the same tracer particle speeds, and (iii) have zero dispersion of tracer particles regardless of length of flow. It is hard to conceive of a more contrived and unrealistic flow scheme. The transit time distributions in stream tube models are actually quite complex because of the combined effect of varying stream tube length distributions and varying dispersion characteristics (Bardsley, 2003).

With respect to the aquifer whose permeability "increases systematically with depth", this represents an entirely undefined entity. What is the nature of the "systematic" increase in permeability that will give rise to a mathematical derivation of an exponential transit time distribution for an aquifer? No reference is offered in support of the assertion and it is worth noting that a strong hydrological argument has been made that exponential transit time distributions are not to be expected from aquifers (Kirchner, 2017).

With reference to "The statement about gamma distributions is likewise misplaced", what is the particular statement that is supposedly misplaced? There seems no argument with the Nash Cascade gamma model – which gives of course integer values of the gamma shape parameter because any sum is composed of an integer number of components $\geq$ 1.

Further aspects of the gamma distribution are also cited by the reviewer:

*In contrast, the empirically determined shape factors among the 20 catchments analyzed by Godsey et al. (2010) ranged from about 0.3 to 0.8. None were even close to 1. Approximate gamma distributions with shape factors near 0.5, consistent with the Godsey et al. findings, can potentially arise from advection and dispersion with spatially distributed inputs (Kirchener et al. 2001).*

The Godsey et al (2010) authors elected to fit gamma distributions to transit time data and obtained gamma distribution shape parameters from the fitting process in the range of about 0.3 to 0.8. The reviewer seems to attach much significance to these gamma shape parameter values as somehow supporting his assertion that my offending statement (which is presumably "gamma distributions do not warrant special consideration as transit-time distributions outside of idealised situations") is misplaced.
In fact, whether gamma distribution shape parameters as obtained from fitting to data are much less than 1, near to 1, or much greater than 1 is irrelevant to any argument seeking to justify the gamma distribution. The cited reference (Kirchner et al. 2001) deals with a special case where a perfectly uniform slope segment with uniform tracer increments along its length gives rise to a particular form of mixed inverse Gaussian transit time distribution which has similarities to the gamma distribution with shape parameter < 1. Even then the gamma model only holds for the special case of this special case where advection and dispersion are of roughly equal effectiveness in transporting the tracer to the stream. To upscale from such an ultra-idealised local model to argue that there is therefore theoretical justification for real-world catchments to have gamma-distributed transit times (of whatever shape parameter) is simply unrealistic. On the other hand, what can be usefully taken from the data of the Godsey et al (2010) paper is that catchment transit time distributions will often be L-shaped, though certainly not necessarily gamma-distributed. Given that a gamma shape parameter range of 0.3-0.8 corresponds to a skewness range of 2.2-3.7, there is no reason why any L-shaped probability distribution within this skewness range could not serve as well to describe the data.

As an aside unrelated to the paper, in the case of groundwater systems the usual approach to making first estimates of transit time distributions for groundwater systems would be by utilising particle tracking options in numerical groundwater models. It is surprising then that there should have been such longevity of analytical approaches to transit time distributions in the arguably more complex case of catchment hydrology. Perhaps there is scope for agent-based numerical catchment modelling to provide a means of simulating catchment transit time distributions.

The previous negative comments concerning parametric distributions do not of course imply that the nonparametric lower bound approach has value. That remains to be determined by application to real data. There is also clearly a need for any revised version of the paper to include an L-shaped transit time distribution, as noted by the reviewers.

*Sections 2 and 3 (definitions and steady-state case):*
*From a purely theoretical standpoint I do not see any problems here, but I am not an*
*expert in linear programming techniques. From a practical standpoint the steady-state*
*case is relevant only as a (potentially rather poor) approximation to the real world.*
*Section 4 (non-steady-state case):*
*The time-varying case presented here, in which the shape of the transit time distribution can change but its mean*
*must stay the same, is inconsistent with the entire*
*literature on catchment nonstationarity. Even theoretically, it is very difficult to imagine*
*any catchment that could possibly work this way. This section may be interesting from*
*a mathematical standpoint but is completely irrelevant to the real world.*

*The gamma PDF's (with shape factor of 5) that are used to generate the test time*
*series bear no resemblance to real-world catchment PDF's (with shape factor well below 1). There is no reason to*
*assume that a method that works with such an unrealistic*
*test time series will necessarily work with a more realistic one.*

Other reviewers have made the same points. The steady state case was only included as it has been around for some time and, being a simple case, seemed a logical means to illustrate the linear programming approach. Similarly, the constant mean with varying distributions was just the next level of complexity up from that. Given reviewer comments, any revised paper would be best to focus just on nonstationary L-shaped transit time distributions with different forms and different means and see how they go. Obviously the lower bound approach needs to be robust enough to handle L-shaped transit time distributions if it is to have practical value.

*The nonparametric TTD is unrealistically truncated at 23 months. Of course, given*
*that the generating distributions – see (a) above – have trivial tails beyond 23 months,*
*the truncation effects are small. But in the real world, where one cannot know this*
*in advance, what could be the justification for not extending the nonparametric TTD*
*to much longer lags? In that case, of course, the computations would become more*
*difficult and, more importantly, the solutions would become much less constrained.*

It has to be remembered that a lower bound to the mean is being sought and not the mean itself. The critical issue is not whether the true transit time distribution has a long tail beyond the upper bound to the nonparametric distribution, but whether the choice of upper bound will have any influence on determining the value of the lower bound to the transit time mean (for a given degree of data matching). This can only be determined after the minimisation process. This is noted also on p.9 (line 30). As can be seen from Fig 5 (c) (d), the lower bound to the mean would have been the same value if the nonparametric upper bound had been set to, say, 2,300 months instead of 23 months (but at the expense of much more calculation). These solutions are therefore not less constrained by increasing the upper bound to the nonparametric distribution.

*The truncation of the nonparametric TTD automatically imposes bounds on the mean*
*– between 0 and 23 months in theory, but in practice, with any dispersion (either physical or numerical), the range*
*will be narrower and the tendency will be more toward the*
*center. Thus it is guaranteed that the solution will not be that far from 6 or 12 months,*
*which are known in advance to be the correct answers.*

The tendency will certainly not be more toward the centre in general. As noted on page 9 (line 7) the single gamma transit time distribution behind the data was well-discovered because of the linear nature of the data simulation process, not because of some tendency toward the centre. In any revised version of the paper some long-tailed distributions will be included to illustrate that the lower bound to the mean may sometimes be much less than the true mean transit time.

*The simulated time series are unrealistically long (not many sites have 21 years of tracer data).*

A good point. An example with a shorter simulated time series should be included in a revision.

*The simulated time series are unrealistically PERFECT. There are no measurement errors in either the input or output time series. Random numbers are added to the square wave in order to create the X values, but then the Y values are convolved with zero error, and then the X and Y values are used – with zero error – in the proposed LP inversion technique. This bears no resemblance to the real-world inversion problem, where both the inputs and outputs will have errors. Furthermore, those errors are potentially catastrophic for an poorly constrained inversion technique like the proposed LP method*

In a revised version a measurement error will be added to both the input and output time series. It isn't obvious though that adding errors will have a "catastrophic effect" effect provided the errors are not too large. It will be interesting to see the outcome.

*In summary, is such a contrived test case that it gives no useful information about the robustness or the practical utility of the proposed method. There is no guarantee that, in a more realistic test case, the proposed method would give a meaningful lower bound estimator. Indeed, the result could potentially even exceed the true mean.*

As noted above, the proposed more realistic example should clarify such issues – at least to the extent that, if successful, others may be encouraged to try out the method.

**References**

Bardsley, W.E. 2003. Temporal moments of a tracer pulse in a perfectly parallel flow system *Advances in Water Resources* 26, 599–607.

Kirchner, J. 2017. Interactive comment on "Aggregation effects on tritium-based mean transit times and young water fractions in spatially heterogeneous catchments and groundwater systems, and implications for past and future applications of tritium" by M. K. Stewart et al. Hydrol. Earth Syst. Sci. Discuss., doi:10.5194/hess-2016-532-SC4, 2017

---

## Author Comment (AC4) · 27 May 2017

Reply to reviewer # 3

My thanks for the comments – which appear below in italics and responses follow in plain text.

*1. relevance. As noted by other reviewers, the practical and/or theoretical relevance of this paper is unclear to me. Getting lower bonds for Mean Transit Times does not seem to be an obviously relevant scientific problem; therefore, the author should better discuss the important implications of this work for hydrology and interpretation of environmental time-series. Interestingly, the paper by Jim Kirchner who apparently inspired this work, suggested that the mean transit time is a poor representation of this type of systems (as noted by the author in the introduction). In fact, tracer dynamics are mostly driven by the fraction of young water, and its time-variability.*

Getting lower bounds to mean transit times can certainly be of practical relevance for catchment situations, given a high lower bound can be established. This indicates the existence of a large mean transit time (whatever its true value may be), which can have implications for how the catchment interacts with its environment:

Additionally, catchments with long transit times are more likely to be resilient to short-term (years to decades) variations in rainfall but will respond to climate or land use changes that cause longer-term (decades to centuries) changes in groundwater recharge and flow.
Cartwright and Morgenstern (2016).

On smaller time scales, establishing that a mean transit time exceeds even a few months may give indication that any contaminant released into the catchment may take some time to flush itself out.

However, it is not for me nor the reviewers to make a judgement call on the value of the concept of lower bounds to mean transit times. It should be left for users to try it out for themselves and see if they find if this approach has use for them.

It is of course desirable to estimate part or all of a transit time distribution but this technical note is just concerned with the possibility of quickly gaining at least some information of catchment storage behaviour via a nonparametric approach with minimal assumptions.

The fraction of young water does have some connection to the lower bound in that a small proportion of young water would be expected to result in a higher lower bound for the mean transit time.

*2. tools. there are several papers that have shown unambiguously why the lumped convolution approach is misleading. It is quite surprising to see that the lumped convolution approach is still used (in this case in a discrete fashion). Therefore, in my view section 3 is completely useless - as that lumped formulation is not able to describe real-wold catchment dynamics. Note that Kirchner [2016a] used that approach only to explain the idea of aggregation bias in simple terms. Here the steady nature of TTDs (or MTTS) is a fundamental assumption (in fact, all the numerical examples shown in the paper refer to steady state TTDs).*

It should not be so surprising for the reviewer to see that the lumped convolution approach is still used. See in this journal, for example, Mosquera et al. (2016), who no doubt will consider that their work is not useless. That paper also outlines some practical difficulties involved with using time-varying models.

However, the lower bound paper is not in any case concerned specifically with the steady state condition. It is therefore confusing to see the reviewer's reference to time-invariant transit time distributions supposedly being a fundamental assumption. Section 3 was titled as being for the case of the steady state situation, which by definition has the steady state assumption. The rest of the

lower bound paper is concerned with conditions other than steady state, going on to the next level of complexity (time-varying transit time distributions with constant means), with some brief comment at the end of Section 4 about the more general situation of time-varying means.

In hindsight, it certainly would have been better to have considered only the most general case with time-varying means. A revised version of the paper would take that approach. Also, the example does need to be an illustration of the general time-varying case.

*3. tools. I have several problems also with the time varying version of the approach (section 4). first of all, the author apparently uses "forward distributions" and not "backword distributions" in equation (5). If this is correct, then the convolution in eq. (5) should refer to mass fluxes and not to concentrations, as in section 3 (see also the periodic nature of the input in Figure 2). Overall, the insisted use of the terminology "tracer time-series" is confusing. The meaning (and the nature) of the kernel linking input-output signals in environmental systems changes, depending on the quantity involved (concentrations, mass fluxes, water fluxes), as extensively discussed in the literature of the 70s by Niemi, Zuber and many others.*

The method is only applicable with backward distributions (predicting the output time series at the observation point at a given time). It may be the arrangement of the time ordering indexing gave the impression of forward distributions. All the time series are defined to be flux weighted (product of flux and concentration), and not concentrations (line 26, page 2). The driving force of the flux variation is irrelevant to the LP methodology, which is only concerned with tracer particle fluxes and not their causes.

*Moreover, the constraint that the mean of the time-varying distributions is the same (Section 4) is untenable. There are tens of experimental and theoretical studies that show unambiguously how the mean travel time changes depending on hydrologic conditions (e.g. catchment wetness).*

It was only an example of the next level up of LP complexity. It was incorrectly specified in the paper in any case and there are no problems in removing it, along with the steady state model.

*4. results. Based on my previous comment I find quite surprising that all the numerical examples refer to the steady state system.*

Yes .. all examples need to be for time-varying general nonparametric models.

*5. as noted by other reviewers, the referencing is definitely inappropriate. I think the author is missing a huge number of papers about travel time formulation and application. As an example, the references to Kirchaner, Selle and Peralta-Tapia whan talking about TTD time-variance should be properly complemented by those works where the idea of time-variant TTDS has been proposed, proved theoretically and then applied. The relevant missing papers are too much to list them here.*

As noted in response to other reviewers on this topic, this short technical note is just concerned with proposing a methodology for combining linear programming with nonparametric transit time distributions to obtain a lower bound to mean transit time. To balloon out the text to incorporate a massive introduction covering all that has gone before in parametric transit time modelling would be a needless distraction and go far beyond the size and scope of a short technical note with narrow focus. However, because a revised paper will now be themed only on the time-varying general model, it would certainly be appropriate to trace the initiation and subsequent development of time-variability of transit time distributions. Part of this would need to overview the extent to which means of sequential transit time distributions may differ, as this can be set up as an LP constraint when obtaining the lower bounds.

*Overall my impression is that this work is a nice mathematical exercise that unfortunately disregards the physical processes involved in catchment transport processes, and (as such) it as a reduced potential for real world hydrological applications.*

I guess "nice" is being used here in the context of "irrelevant". However, the reviewer's main issue appears to have been that the nonparametric lower bound approach supposedly does not allow for full degrees of freedom for time-variability of transit time distributions. Hopefully this concern could be offset by considering in a revised paper only the general time-varying case, together with examples.

Taking into account all reviewer comments, the obvious synthetic example to use in a revised paper would be for a sequence of different L-shaped distributions with variations of both mean and shape, together with input and output time series including a component of measurement error. By "L-shaped" is meant (for a probability density function) that the first derivative is everywhere ≤ 0 and the second derivative is everywhere ≥ 0. This includes, for example, all gamma distributions with shape parameter ≤ 1. The required linear constraints would be applied to all the different time-varying nonparametric transit time discrete distributions. This would create a much larger LP setup than was used in the steady state example, but is certainly achievable for current LP packages.

Cartwright, I, Morgenstern, U, 2016. Contrasting transit times of water from peatlands and eucalypt forests in the Australian Alps determined by tritium: implications for vulnerability and the source of water in upland catchments. *Hydrol. Earth Syst. Sci.*, 20, 4757–4773.

Mosquera, G. M. et al., 2016. Insights into the water mean transit time in a high-elevation tropical ecosystem. *Hydrol. Earth Syst. Sci.*, 20, 2987–3004.